# Thermo-Mechano-Chemical Processing of Printed Circuit Boards for Organic Fraction Removal

Sergey M. Frolov [1,*], Viktor A. Smetanyuk [1], Anton S. Silantiev [1], Ilias A. Sadykov [1], Fedor S. Frolov [1], Jaroslav K. Hasiak [2], Alexey A. Shiryaev [3] and Vladimir E. Sitnikov [3]

[1] Department of Combustion and Explosion, N. N. Semenov Federal Research Center for Chemical Physics, Moscow 119991, Russia; smetanuk@mail.ru (V.A.S.); silantevu@mail.ru (A.S.S.); ilsadykov@mail.ru (I.A.S.); f.frolov@chph.ru (F.S.F.)

[2] A. N. Nesmeyanov Institute of Organoelement Compounds of Russian Academy of Sciences, Moscow 119991, Russia; hasiak1996@gmail.com

[3] JSC South Ural Specialized Recycling Center, Miass 456313, Russia; shiryaev@centrutil.ru (A.A.S.); sitnikov_ve@centrutil.ru (V.E.S.)

* Correspondence: smfrol@chph.ras.ru

**Abstract:** Printed circuit boards (PCBs) are the main components of e-waste. In order to reduce the negative impact of waste PCBs on human health and the environment, they must be properly disposed of. A new method is demonstrated for recycling waste PCBs. It is referred to as the high-temperature thermo-mechano-chemical gasification (TMCG) of PCBs by the detonation-born gasification agent (GA), which is a blend of $H_2O$ and $CO_2$ heated to a temperature above 2000 °C. The GA is produced in a pulsed detonation gun (PDG) operating on a near-stoichiometric methane–oxygen mixture. The PDG operates in a pulsed mode producing pulsed supersonic jets of GA and pulsed shock waves possessing a huge destructive power. When the PDG is attached to a compact flow reactor filled with waste PCBs, the PCBs are subject to the intense thermo-mechano-chemical action of both strong shock waves and high-temperature supersonic jets of GA in powerful vortical structures established in the flow reactor. The shock waves grind waste PCBs into fine particles, which undergo repeated involvement and gasification in the high-temperature vortical structures of the GA. Demonstration experiments show full (above 98%) gasification of the 1 kg batch of organic matter in a setup operation time of less than 350 s. The gaseous products of PCB gasification are mainly composed of $CO_2$, $CO$, $H_2$, $N_2$, and $CH_4$, with the share of flammable gas components reaching about 45 vol%. The solid residues appear in the form of fine powder with visible metal inclusions of different sizes. All particles in the powder freed from the visible metal inclusions possess a size less than 300–400 µm, including a large fraction of sizes less than 100 µm. The powder contains Sn, Pb, Cu, Ni, Fe, In, Cd, Zn, Ca, Si, Al, Ti, Ni, and Cl. Among these substances, Sn (10–20 wt%), Pb (5–10 wt%), and Cu (up to 1.5 wt%) are detected in the maximum amounts. In the powder submitted for analysis, precious elements Ag, Au, and Pt are not detected. Some solid mass (about 20 wt% of the processed PCBs) is removed from the flow reactor with the escaping gas and is partly (about 10 wt%) trapped by the cyclones in the exhaust cleaning system. Metal inclusions of all visible sizes accumulate only in the flow reactor and are not detected in powder samples extracted from the cyclones. The gasification degree of the solid residues extracted from the cyclones ranges from 76 to 91 wt%, i.e., they are gasified only partly. This problem will be eliminated in future work.

**Keywords:** printed circuit boards; precious metals; organic fraction removal; high-temperature $H_2O/CO_2$ gasification; pulsed detonation gun; syngas; solid residue particles

## 1. Introduction

Electronic waste (e-waste) is a type of waste containing electronic and other electrical devices and their metal and nonmetal parts. The volume of e-waste in municipal solid waste (MSW) is gradually growing [1,2]. Printed circuit boards (PCBs) are the main

components of e-waste. Despite the fact that their share in the total volume of MSW is relatively small, they contain a significant amount of hazardous and toxic pollutants due to such substances as mercury and the components of polymer substrates (resin), glass fiber epoxy laminates, and fill compounds like polychlorinated biphenyls, polybrominated biphenyls, polyvinyl chloride, brominated flame retardants, polybrominated diphenyl ethers, and tetrabromobisphenol-A [3–5]. Some of them cause the formation of dioxins and furans during e-waste smelting and incineration [6,7]. Parts of PCBs, connectors and wires, capacitors, and batteries contain inclusions of cadmium, nickel, arsenic, copper, lead, antimony, asbestos, etc. Electronic waste includes valuable, rare earth, and precious metals that can be reused. The main recyclable material is iron, and the main non-ferrous metals are aluminum and copper. The content of precious metals, e.g., gold, silver, platinum, and palladium, may exceed the content of these metals in the original ore during their extraction. It is important to extract useful components for their reuse and return them to the production cycle after appropriate preparation.

In order to reduce the negative impact of waste PCBs on human health and the environment, they must be properly disposed of. There exist several approaches to PCB recycling. One approach is simply landfilling [8–10]. It is reported in [2,11,12] that most of the nonmetal (organic) wastes of PCBs (76–94%) are treated as landfill, while about 40% go to uncontrolled landfilling. Another approach is based on the mechanical processing of PCBs, which includes disassembly, crushing, pulverizing, and separation [13–16] for liberating various metals from cladding materials such as resin, fiberglass, and plastics. For this purpose, different energy-intensive crushers (hammer, rotary, disk, etc.), mills (disk, ball, etc.), cutters, and shredders with sieves are used. An alternative to landfilling and mechanical processing is the thermal processing of waste PCBs via open burning [17], smelting [18], incineration [19,20], low-temperature (up to 800 °C) pyrolysis [21,22] and autothermal/allothermal gasification [23–26]. Open burning, smelting and incineration are environmentally unsafe due to emissions of hazardous and toxic pollutants. During low-temperature pyrolysis, the organic matter is decomposed to the syngas containing noncondensable and hazardous condensable (tar) gaseous hydrocarbons and phenolic tar, whereas solid residues, including char, slag, and minerals are further processed to extract metals. The gasification of waste PCBs involves partial oxidation of organic constituents with the aid of an externally fed gasifying agent (GA) containing either free or bound oxygen (air, $O_2$, $H_2O$, $CO_2$) to produce syngas. The maximum conversion efficiency is achieved if all carbon is oxidized to CO. Despite steam and $CO_2$-assisted gasification currently being considered the most sustainable and effective method for waste PCB management, the thermal processing of e-waste is still treated as environmentally unsafe due to emissions of the various pollutants such as HCl, SOx, HF, NOx, as well as tar and char [20]. Other approaches to PCB recycling include hydrometallurgical [27,28] and pyrometallurgical [29] methods. The hydrometallurgical method is a combination of mechanical refining, leaching, separation–purification, and metal recovery. It involves the production of a significant amount of liquid waste and sludge. The pyrometallurgical method is the combination of incineration, smelting, dressing, sintering, melting, and high-temperature gas-phase reactions in a furnace. The stages of incineration and smelting generate hazardous emissions such as dioxins, furans, halogens, and volatile metals.

There are several methods to identify and evaluate the distribution, composition, morphology, and phases of different components in PCBs [30–32], including optical microscope analysis, synchrotron X-ray tomography, X-ray diffraction (XRD), scanning electronic microscopy/energy dispersive spectroscopy, and inductively coupled plasma spectroscopy. The uncertainty in determining all these properties is known to be strongly dependent on the particle size and sample mass chosen for the analysis [31]. The reduction of particle size is an important pre-requisite for the subsequent effective extraction of metals from associated nonmetal constituents in PCBs. It is achieved by various mechanical means like energy-intensive milling [33] and even cryo-milling [34] involving losses of precious metals in fine fractions. For analyzing the composition of pyrolysis products, gas chromatography

(GC) is used. Fixed carbon, volatile matter, and ash contents of the original PCB and solid residues of its pyrolysis are usually determined by the CHNS/O analyzers [35,36].

In this paper, we apply a new method for recycling waste PCBs, which is referred to as the high-temperature thermo-mechano-chemical gasification (TMCG) of PCBs by the detonation-born GA, that is the ultra-superheated mixture of $H_2O$ and $CO_2$. The detonation-born GA with a temperature exceeding 2000 °C is produced in a pulsed detonation gun (PDG) operating on any available gaseous or liquid hydrocarbon fuel using gaseous oxygen or air as oxidizer [37]. In the detonation gun, the fuel–oxidizer mixture is converted to the high-temperature combustion products (mainly $H_2O$ and $CO_2$) due to overall exothermic self-accelerating chemical reactions induced by volumetric compression and heating in a strong, self-sustaining shock wave (SW). The PDG operates in a pulsed mode producing pulsed supersonic jets of GA and pulsed SWs possessing a huge destructive power. When the PDG is attached to a compact flow reactor filled with waste PCBs, the PCBs are subject to intense thermo-mechano-chemical action of both strong SWs and high-temperature supersonic jets of GA in powerful vortical structures established in the flow reactor [38]. The SWs grind waste PCBs into fine particles, which can undergo multiple acts of fragmentation by the successive incident and reflected SWs, as well as undergo repeated involvement and gasification in the high-temperature vortical structures of the GA. The ability of the GA composed of $H_2O$ and $CO_2$ to gasify organic wastes, producing no negative effects on the environment, is well known [39]. At temperatures above 1500 °C, tar and char formed at the initial stages of the gasification process are completely transformed into syngas, ideally composed only of $H_2$ and CO in a proportion depending on the feedstock, whereas condensed mineral residues consist of safe, simple oxides and aqueous solutions of oxygen-free acids, such as HCl, HF, and $H_2S$, and ammonia $NH_3$.

The TMCG method has already been successfully demonstrated for the natural gas conversion [40] and gasification of liquid/solid wastes (waste machine oil, sawdust, sunflower seed husks, etc.) [40–43]. The objective of this paper is to demonstrate the TMCG method on the gasification of waste PCBs using a new experimental setup and accompanying measurements of gaseous and solid gasification products by the chromatography–mass spectrometry (GC-MS), CHN analysis, wet laser diffraction method, and X-ray fluorescence (XRF).

## 2. Materials and Methods

Figures 1 and 2 show the schematic and photographs of the experimental setup. The setup consists of a 40 L volume vertical flow reactor, a PDG, and an exhaust cleaning system. The PDG is attached tangentially to the flow reactor at its bottom. The PDG is a tube with an internal diameter of 50 mm, a length of 900 mm, and a volume of 1.8 L, equipped with a mixing and ignition device, a cooling jacket, and ports for ionization probes (IPs) used for measuring the detonation propagation velocity. A near-stoichiometric natural gas–oxygen mixture with a fuel-to-oxygen equivalence ratio of $0.92 \pm 0.03$ is used as a combustible gas for the PDG and as a source for high-temperature GA. The average mixture flow rate through the PDG is $3.2 \pm 0.2$ g/s. A batch of feedstock (waste PCBs) is loaded into the flow reactor through the top hatch. The solid residue is unloaded through the lower hatch. Gases are removed from the flow reactor through the central channel, passing through the top hatch with a flow section diameter of 15 mm. The outlet channel is recessed relative to the top hatch by 100 mm. There are also ports for a low-frequency pressure sensor Kurant-DA 1.6 MPa (Russia) and N-type thermocouple Owen DTPN286M –40/+1250C (Russia) on the top of the flow reactor.

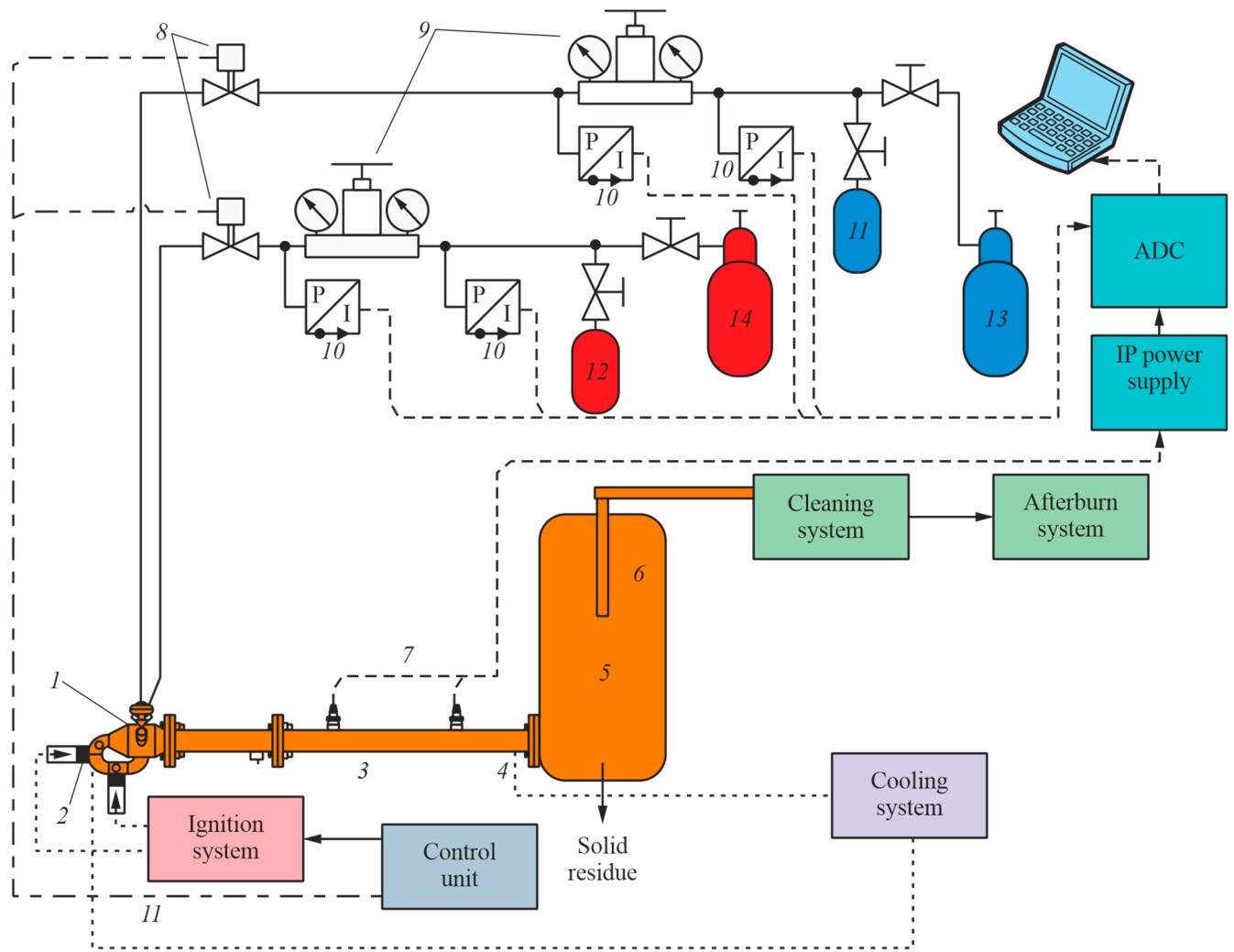

**Figure 1.** Schematic of the experimental setup: 1 is the mixing and ignition device, 2 are the spark plugs, 3 is the PDG, 4 is the cooling system, 5 is the flow reactor, 6 is the sampling system for gaseous products, 7 is the ionization probes (IPs), 8 is the oxygen and fuel line valves, 9 is the reducers, 10 is the pressure sensors, 11 is the oxygen receiver, 12 is the natural gas receiver, 13 is the oxygen cylinder, 14 is the natural gas cylinder.

Gas composition is measured by the flow gas analyzer MRU VARIO SYNGAS PLUS (Germany) and by Chromatec-Crystal 2000 (Russia) and GC-MS Chromatec-Crystal 5000 (Russia) gas chromatographs. The granulometric analysis of the ash powder in the solid residue is performed by the wet laser diffraction method using Analysette 22 device (Fritch, Idar-Oberstein, Germany). The elemental composition of the solid residue is determined by CHN analysis using CHNS/O analyzer (Vario EL cube, Langenselbold, Germany) with accuracy of 0.30 vol% abs. Qualitative and quantitative analyses of solid residue samples are carried out by the XRF technique using SPECTROSCAN MAX-GVM (Russia).

The exhaust gas cleaning system consists of two 2 L identical cyclones (cyclones #1 and #2 in Figure 2b) separated by three water-cooled sections for cooling the gasification products. The cyclones are designed for centrifugal separation of solid particles and do not contain filtering elements. At the end of the cleaning system there is an ejection-type burner for burning flammable gasification products and visual monitoring of the process. The burner is equipped with a pilot flame system so that the gasification products can be readily ignited and burned in a torch. The burning time of the torch on the burner is treated as the characteristic time of the TMCG of organic compounds in waste PCBs.

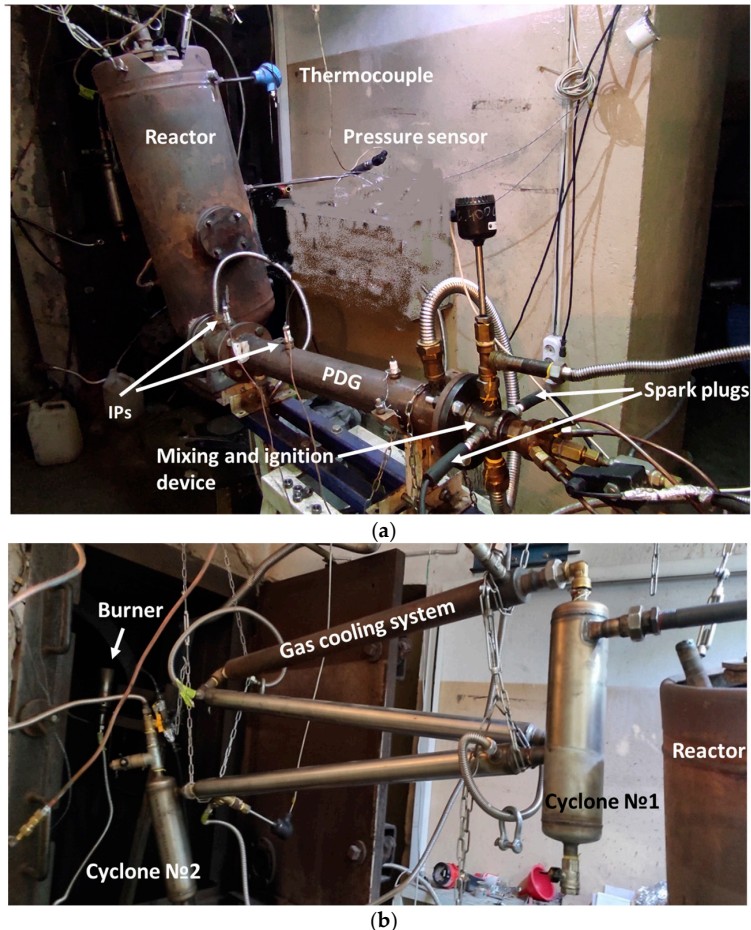

(a)

(b)

**Figure 2.** Photographs of the experimental setup (**a**) and the cleaning system of gasification products (**b**).

The feedstock consisted of waste PCBs of FR-2 type cut into pieces ranging in size from 10 × 10 mm to 30 × 30 mm with various attached electronic components (Figure 3). The basis of PCB is textolite consisting of layers of fiberglass and an adhesive (compound) base, the complete removal of which is the goal of the work. The elemental composition of waste PCBs was not analyzed.

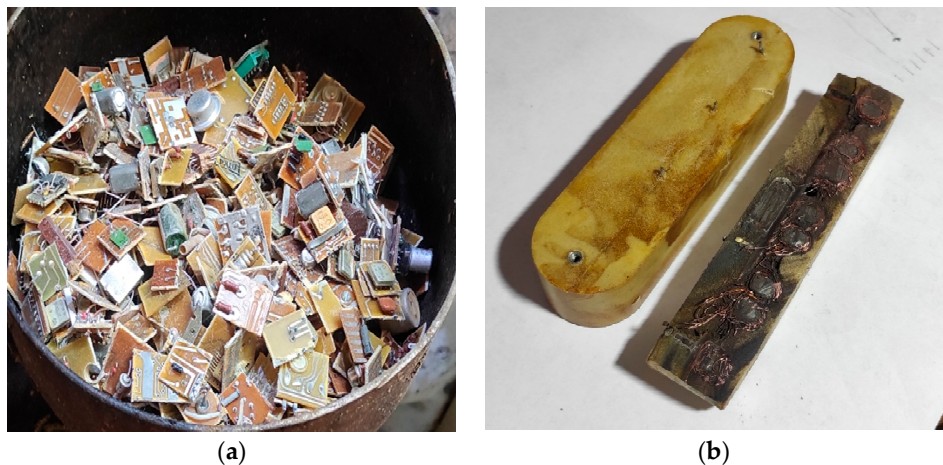

(a) (b)

**Figure 3.** The feed e-waste: (**a**) PCBs and (**b**) PCBs potted with an organic compound.

The detonation velocity in the PDG is determined based on the recordings of two IPs (see Figures 1 and 2). Figure 4 shows the typical recordings of the IPs in 14 successive cycles of the PDG and the exploded view of the recordings in a single cycle. Based on such recordings, we determine the detonation velocity as the ratio of the known distance between the IPs ($\Delta L = 250$ mm) to the time interval $\Delta t$ between signals, $D = \Delta L / \Delta t$. In experiments, the measured detonation velocity in the natural gas–oxygen mixture is $2100 \pm 100$ m/s. Table 1 shows the composition of the natural gas consisting mainly (96.1 vol%) of methane. Table 2 shows the composition of the GA—the mixture of ultra-superheated $H_2O$ and $CO_2$—measured in the experiments without supply of waste PCBs using the flow gas analyzer MRU VARIO SYNGAS PLUS (Germany); the measurement error is estimated at 5%.

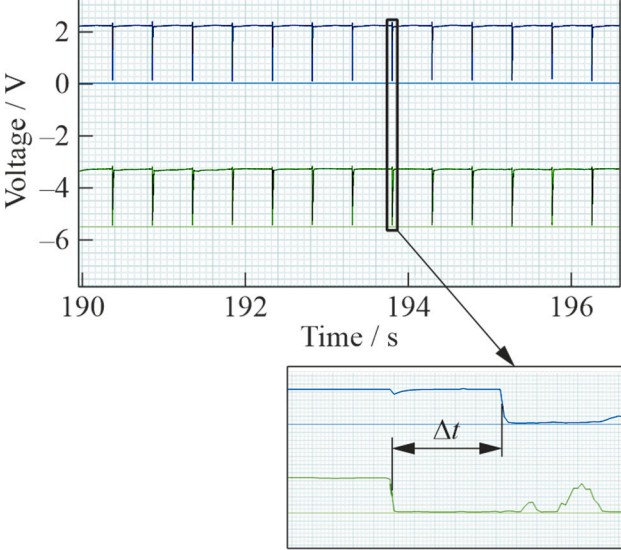

**Figure 4.** Recordings of ionization probes during setup operation.

**Table 1.** Compositions of natural gas.

| Species | vol% |
| --- | --- |
| $CH_4$ | 96.1 |
| $C_2H_6$ | 2.1 |
| $C_3H_8$ | 0.6 |
| $C_4H_{10}$ | 0.2 |
| $N_2$ | 1.0 |

**Table 2.** Measured composition of the gasifying agent.

| Species | vol% |
| --- | --- |
| $H_2O$ [1] | 65.2 |
| $CO_2$ | 31.8 |
| CO | 1.9 |
| $H_2$ | 1.1 |

[1] by difference.

The PCBs are processed according to the following algorithm. Firstly, a batch of PCBs (1 kg) is loaded into the flow reactor. Then, the data acquisition system, as well as the exhaust cooling system and the PDG, are activated. Thereafter, the cyclic operation of the PDG is started with a preset frequency. In the flow reactor, with the cyclic generation of strong SWs and the injection of high-temperature GA, the PCBs are heated, fragmented, and gasified while formed heavy and light particles of the solid residue are spatially separated

in the intense vortical flow. Due to differences in density and mass, particles of different sizes settle at different speeds and occupy their own areas along the flow reactor height. Thus, the flow reactor partially performs the functions of a cyclone when solid particles move along the walls, and gas is taken from the center of the vortex and escapes from the flow reactor through the recessed outlet channel. When the mean temperature in the upper part of the flow reactor reaches 200 °C (in about 20 s from the start of setup operation), a flame torch appears on the exhaust burner, indicating the beginning of the gasification of organic compounds. The chromatographic probes of the gas escaping from the flow reactor are taken in 40 s from the start of setup operation during the time interval of 20 s. The main gasification phase ends when the flame torch on the burner spontaneously quenches. The processed solid residue is extracted from the reactor and both cyclones in 1 h after feedstock TMCG and analyzed. This time is needed to cool down the flow reactor and cyclones and to dismantle the shut-off elements. In an industrial prototype, this period can be readily reduced to several minutes.

## 3. Results and Discussion

The substrate of PCBs consists of flammable substances with some ash content. To assess the ash content of PCBs, annealing was first carried out with a natural gas–oxygen flame torch. Figure 5 shows the photographs of PCB elements before and after annealing. After annealing 50 g of PCB substrate (textolite) without attached electronic elements, its weight decreases by 15 g, which indicates an ash content of $K_s \approx 70\%$ of pure textolite. Annealing 50 g of some "average" set of PCBs with attached electronic elements shows that their ash content of $K_s \approx 80\%$. Thus, it is expected that during gasification, the mass of feedstock must decrease by 15–20 wt%. If one accounts for possible solid mass removal from the flow reactor with the escaping gas, the total mass loss can be somewhat higher. Thus, annealing experiments show that the initial composition of PCBs includes organic components (15–20 wt%), fiberglass (35–45 wt%), and electronic components (35–50 wt%).

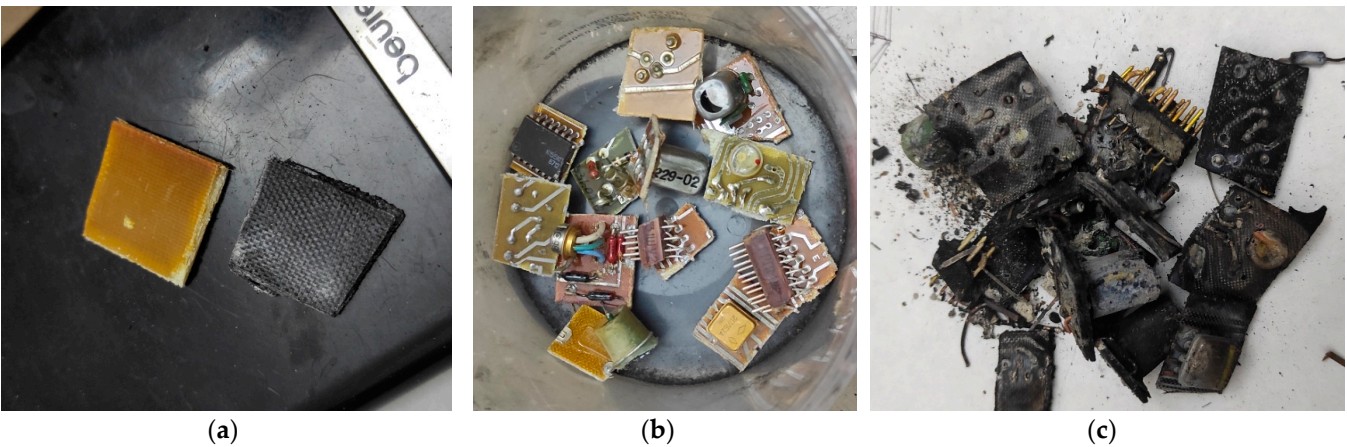

|     (a)     |     (b)     |     (c)     |

**Figure 5.** Elements of PCBs before and after annealing: (**a**) substrate only, (**b**) original substrate with attached electronic elements, and (**c**) processed substrate with attached electronic elements.

It is conditionally assumed that the "full" gasification of organic matter from the waste PCBs is attained when the carbon content in the solid residue of TMCG is less than 2 wt%. In the process of searching for the best operation mode of the gasification setup aimed at attaining full gasification of organic matter, we varied the following parameters: (1) the setup operation time and (2) the PDG operation frequency (1 or 2 Hz).

Figure 6 shows the typical time history of overpressure in the flow reactor measured by the pressure sensor. The overpressure in the flow reactor during setup operation does not usually exceed 0.4 MPa. Figure 7 shows the typical time history of the mean gas temperature in the flow reactor measured by the thermocouple. Note that the measured mean gas temperature should not be treated as the gasification temperature, as gasification

reactions occur inside the flow reactors at local instantaneous temperatures of the GA generated by the PDG. Thus, at the operation frequency of 1 Hz, the local instantaneous maximum temperature of the GA inside the flow reactor during the first quarter of the cycle (~0.25 s) is very high (1500 < $T_{max}$ < 2300 K) [40], while in the second quarter of the cycle, it drops to 1200 < $T_{max}$ < 1500 K and further drops below 1200 K in the second half of the cycle. The drop in the GA temperature is caused by the expansion of high-pressure detonation products behind the detonation wave to the actual pressure in the flow reactor and by mixing the newly produced ultra-superheated GA with the cooled gas in the flow reactor. Calculations show [40] that the time-averaged mean gas temperature in the flow reactor is not much affected by the reactor wall temperature. This means that the walls of the flow reactor can be water-cooled and manufactured from conventional structural materials. In Figures 6 and 7, the gasification time of the PCBs is shown by vertical dashed lines. The time is counted from the beginning of the first PDG operation cycle until the flame torch in the exhaust gas burner is spontaneously quenched. It follows from Figures 6 and 7 that the characteristic time for the experimental setup to reach a quasi-stationary pressure regime in the flow reactor is about 150 s (see Figure 6), while the average gas temperature in the flow reactor is established only after the end of gasification (see Figure 7).

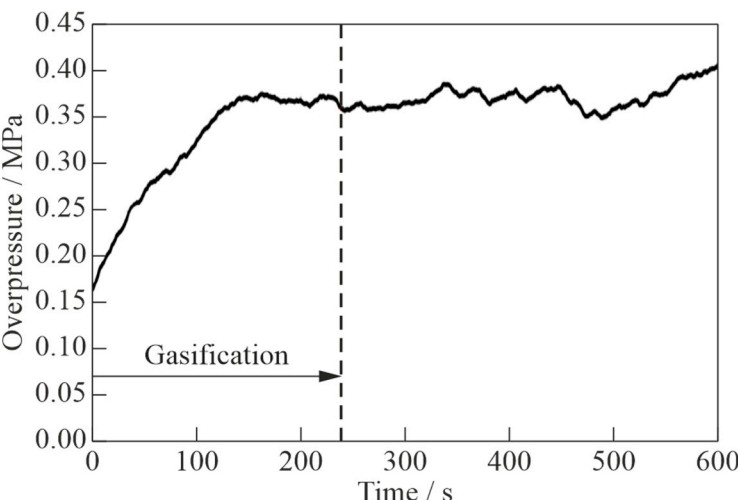

**Figure 6.** Typical time history of the measured mean overpressure in the flow reactor at a PDG operation frequency of 2 Hz.

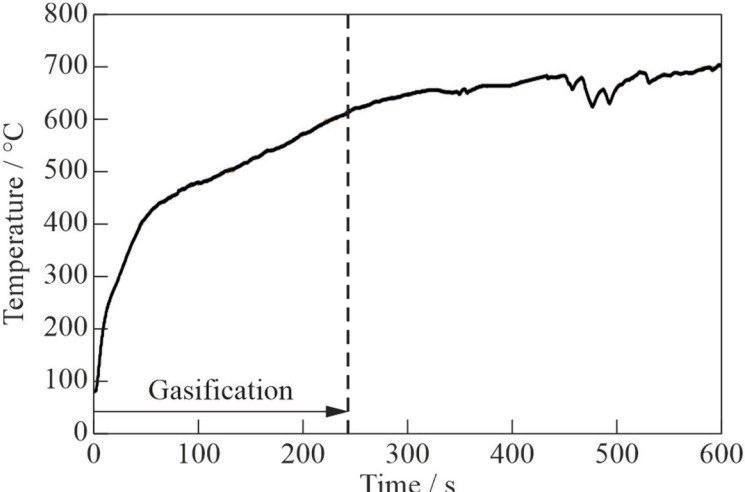

**Figure 7.** Typical time history of the measured gas temperature at the top of flow reactor at a PDG operation frequency of 2 Hz.

Experiments show that the gasification time and the consumption of a combustible methane–oxygen mixture at a PDG frequency of 2 Hz, required for full gasification of organic substances in a 1 kg batch of PCBs, are 240 s and 1.4 kg, respectively. When the PDG operates at a frequency of 1 Hz, full gasification of organic substances in the PCBs requires more time (350 s) but less combustible mixture (about 1.0 kg). Thus, an operation frequency of 1 Hz is preferable for economic reasons but less preferable in terms of performance.

Now, let us consider the properties of the gasification products and solid residues. Figure 8 and Table 3 show a typical GC-MS chromatogram and composition of the gas mixture produced by the gasification of waste PCBs at a PDG frequency of 2 Hz, respectively. The gas is mainly composed of $CO_2$ (52.2 vol%), CO (25.1 vol%), $H_2$ (15.7 vol%), $N_2$ (3.4 vol%), and $CH_4$ (2.3 vol%). The share of flammable gas components is seen to reach about 45 vol%. The qualitative analysis indicates that in addition to the main components listed in Table 3, the gas mixture contains some tar in terms of benzene and its homologs. Figure 9 and Table 4 show a typical GC-MS chromatogram and tar composition in terms of the match with the NIST 20 database, respectively. According to [39], tar must disappear with an increase in the mean gasification temperature, i.e., with an increase in the PDG operation frequency, when the duration of the period of the highest temperatures exceeding 2000 °C increases. In view of this, there is an evident trade-off between economic and ecological considerations: despite the operation frequency of 2 Hz seeming unpreferable compared to 1 Hz from an economic perspective, it must be preferable from an ecological perspective. Obviously, for complete gasification of PCBs, it is necessary to increase the PDG operation frequency.

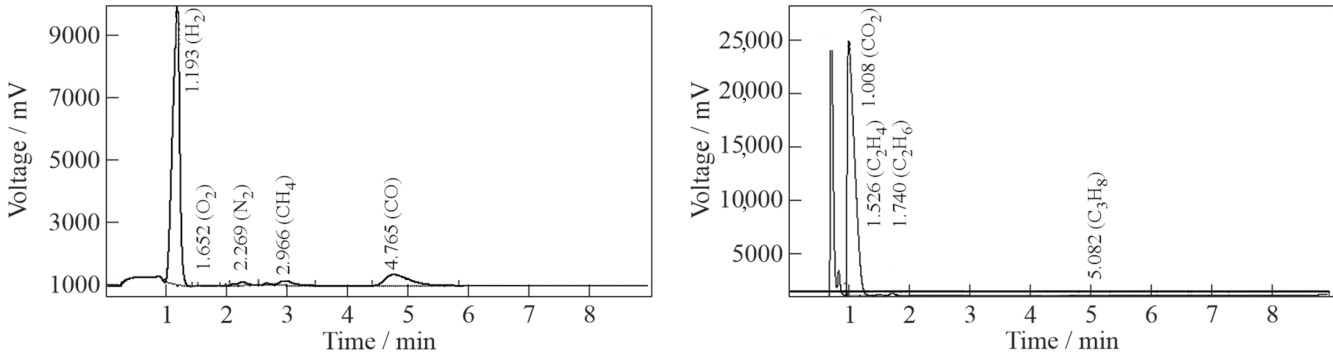

**Figure 8.** A typical chromatogram of the gas mixture produced by the gasification of waste PCBs at a PDG operation frequency of 2 Hz.

**Table 3.** The typical composition of the exhaust gas at a PDG operation frequency of 2 Hz.

| Component | Volume Fraction, vol% Dry |
|---|---|
| $CO_2$ | 52.25 |
| $C_2H_4$ | 0.13 |
| $C_2H_6$ | 0.28 |
| $C_3H_8$ | 0.02 |
| $H_2$ | 15.69 |
| $O_2$ | 0.82 |
| $N_2$ | 3.44 |
| $CH_4$ | 2.25 |
| CO | 25.12 |
| Total | 100.00 |

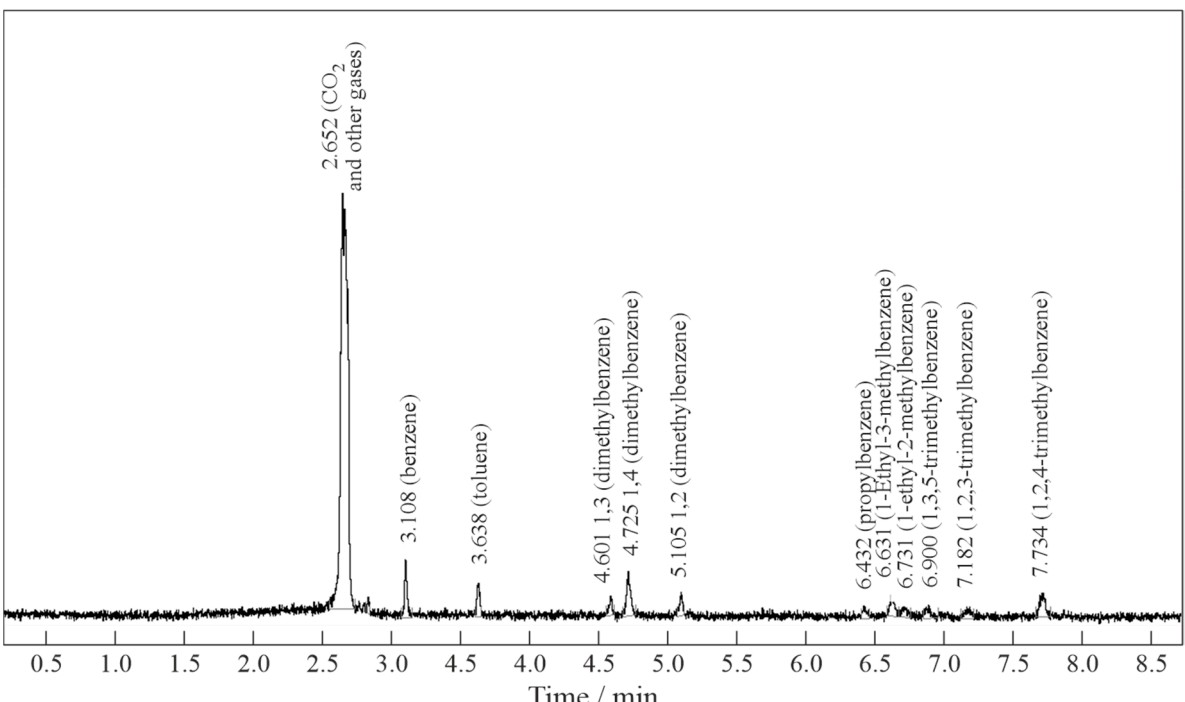

**Figure 9.** A chromatogram indicating the existence of benzene and its homologs in the gas mixture produced by the gasification of waste PCBs in the setup with the PDG operation frequency of 2 Hz.

**Table 4.** Heavy hydrocarbon impurities in the gas mixture produced by the gasification of waste PCBs in the setup at a PDG operation frequency of 2 Hz.

| Impurity | Match with NIST 20 Database |
|:---:|:---:|
| $CO_2$ * | 56.2 |
| benzene | 75.4 |
| toluene | 55.1 |
| 1,3 dimethylbenzene | 67.5 |
| 1,4 dimethylbenzene | 32.5 |
| 1,2 dimethylbenzene | 22.5 |
| propyl benzene | 55.2 |
| 1-ethyl-3-methylbenzene | 31 |
| 1-ethyl-2-methylbenzene | 18.5 |
| 1,3,5-trimethylbenzene | 18.8 |
| 1,2,3-trimethylbenzene | 19.7 |
| 1,2,4-trimethylbenzene | 32.5 |

* $CO_2$ and other gases.

Figure 10 shows the photographs of the solid residues of the gasification of PCBs (Figure 10a) and PCBs potted with an organic compound (Figure 10b). The solid residues are composed of fine powder and metal inclusions. As can be seen, some metal inclusions, which could contain precious metals inside, like body parts, are deformed rather than destroyed. Obviously, to open such body parts, one has to apply a larger-diameter PDG to generate stronger blast waves.

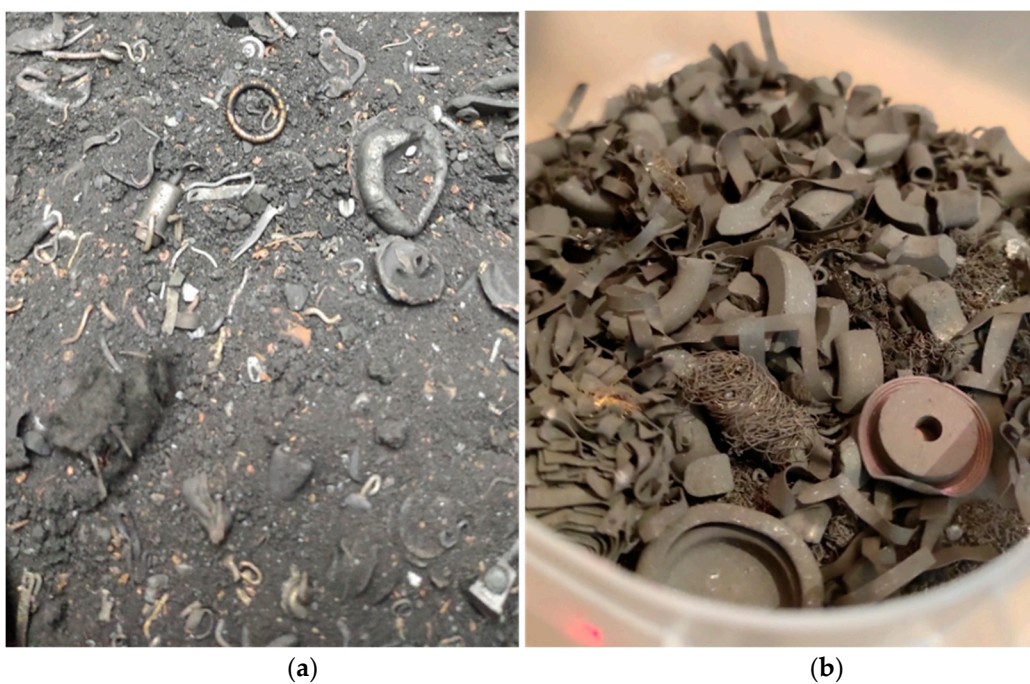

|  |  |
|:---:|:---:|
| (**a**) | (**b**) |

**Figure 10.** Photographs of the solid residues of the gasification of PCBs (**a**) and PCBs filled with an organic compound (**b**).

To obtain the particle size distribution in the powder, the solid residues are first manually freed from visible (macroscopic) metallic inclusions, and then three specimens are taken from different places of each sample and analyzed by wet laser diffraction. Figure 11 presents an example of particle size distributions in the powder. About 30% of all particles possess a size of less than 10 μm, whereas the rest 70% possess a size in the range from 10 to 100 μm. Such a fine powder is the result of the thermo-mechano-chemical action of strong pulsed SWs emanating from the PDG. Table 5 compares particle size distributions for five different samples of PCBs in terms of the average of three measurements for each sample. Samples 1 and 2 are taken from the flow reactor, whereas samples 3, 4, and 5 are taken from the exhaust cleaning system upon the completion of experiments: samples 3 and 4 are taken from cyclone #1, and sample 5 is taken from cyclone #2 (see Figure 2b). The particles of the largest size (200–400 μm) are seen to accumulate in the flow reactor, whereas the smallest particles (20–30 μm) are removed from the flow reactor with the escaping gas and trapped by the cyclones in the exhaust cleaning system. The overwhelming majority (80–100%) of particles in the flow reactor have submillimeter sizes. Metal inclusions of all visible sizes accumulate only in the flow reactor and are not detected in samples extracted from the cyclones. The particles trapped in cyclone #1 appear to be larger than those trapped in cyclone #2: the fraction of the smallest submicron particles in cyclone #2 is a factor of 3–4 larger than in cyclone #1. In general, all particles, extracted from the flow reactor and both cyclones have a submillimeter size. One important circumstance should be noted here: when summing up the material balance, it turns out that a part of the solid mass is removed from the flow reactor with the escaping gas (about 20 wt% of the initial solid mass of the PCBs (fiberglass and electronic components) and only partly (about 10 wt%) captured by the exhaust cleaning system. The fate of the remaining 10 wt% of the solid mass is currently unclear. Apparently, tiny (submicron) particles settle on the internal surfaces of the flow reactor and the exhaust cleaning system and/or are not captured by the cleaning system and enter the flame torch along with the escaping gas. These assumptions will be tested during further work, for example, by installing fine filters upstream of the burner.

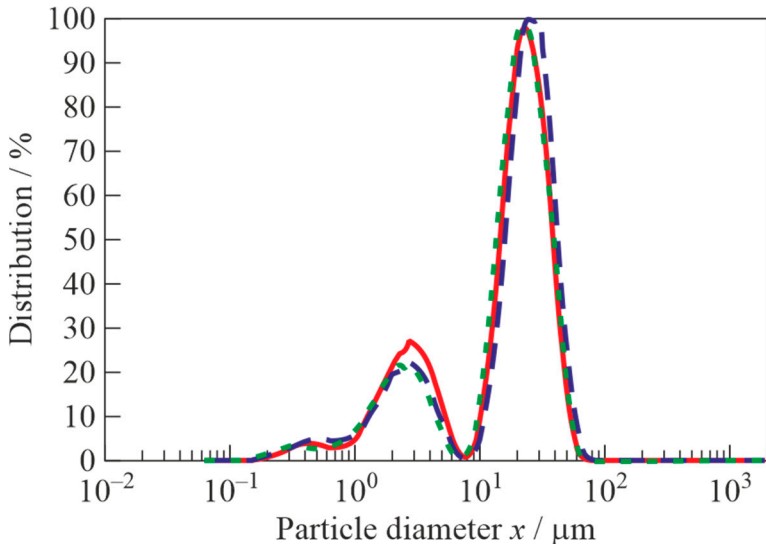

**Figure 11.** An example of particle size distributions obtained by the wet laser diffraction method. The curves correspond to three specimens taken from different places of a powder sample.

**Table 5.** Particle size distributions in the powder for five sets of PCBs after TMCG (%).

| Size | Sample 1 | Sample 2 | Sample 3 | Sample 4 | Sample 5 |
|---|---|---|---|---|---|
| less than 1 μm | 2 | 2 | 5 | 4 | 15 |
| 1–10 μm | 2 | 0 | 25 | 2 | 42 |
| 10–100 μm | 51 | 38 | 70 | 94 | 42 |
| 100–1000 μm | 45 | 60 | 0 | 0 | 1 |
| less than 1 mm | 0 | 0 | 0 | 0 | 0 |
| Max 80–100% | 200–300 μm | 200–400 μm | 20–30 μm | 20–30 μm | (1–5)/(20–60) μm |

Table 6 shows the elemental composition of the powder determined by sample burning in the CHNS/O analyzer. The samples in Table 6 are the same as in Table 5. When burning samples 1 and 2, the remaining ash mass appears to be above 100 wt%. Perhaps the increase in mass occurs due to the oxidation of inorganic components in the powder. In addition, the amount of carbon in samples 1 and 2 turns out to be an order of magnitude lower than in samples 3, 4, and 5, while hydrogen and nitrogen are not detected at all. This indicates the full gasification (over 98 wt%) of organic compounds in samples 1 and 2 after the TMCG of waste PCBs. As for the gasification degree of samples 3, 4, and 5, it ranges from 76 wt% to 91 wt%, while the ash content in these samples ranges from 70 to 90 wt%. These results indicate that the particles removed from the flow reactor with escaping gas are only partly gasified. There are several approaches to solving the problem of increasing the gasification efficiency of organic feedstocks during their TMCG, which are discussed in [37]. One of them is the use of a cascade of communicating flow reactors, allowing gases and particles to move from one flow reactor to another, which increases the average residence time of waste particles in the high-temperature GA and, therefore, increases the gasification efficiency.

**Table 6.** Data on the elemental composition of the powders obtained by the TMCG of waste PCBs.

| Element | Sample 1 | Sample 2 | Sample 3 | Sample 4 | Sample 5 |
|---|---|---|---|---|---|
| C, wt% | 1.59 | 1.80 | 14.42 | 9.35 | 24.18 |
| H, wt% | — | — | 1.24 | 0.30 | 1.39 |
| N, wt% | — | — | 0.48 | — | 0.64 |
| Ash, wt% | (101.00) | (100.70) | 80.55 | 88.20 | 70.73 |
| $\sum$, wt% | 102.59 | 102.50 | 96.69 | 97.85 | 96.94 |

The qualitative analysis of samples presented in Tables 5 and 6 by the XRF method shows that the powders obtained by the TMCG of waste PCBs contain Sn, Pb, Cu, Ni, Fe, In, Cd, Zn, Ca, Si, Al, Ti, Ni, and Cl. It should be reiterated that most of the metals were extracted prior to the analyses, and therefore, their amounts in Table 7 are related only to some residues in the powders. In the powders submitted for analysis, precious elements Ag, Au, and Pt are not detected. This means that these elements (if any) remain in extracted metallic inclusions. Since calcium, silicon, and aluminum are the components of fiberglass, these elements are detected in all samples. The remaining elements are detected as tiny parts of electronic components and solder. Table 7 shows the mineral composition of the powders. The samples are the same as in Tables 5 and 6. As seen, the powders contain tin (10–20 wt%), lead (5–10 wt%), copper (up to 1.5 wt%), and iron (up to 1 wt%). Also present are zinc, indium, cadmium, and nickel.

**Table 7.** Composition of the powders obtained by the TMCG of waste PCBs (wt%).

| Element | Sample 1 | Sample 2 | Sample 3 | Sample 4 | Sample 5 |
|---------|----------|----------|----------|----------|----------|
| Fe | 0.4 | 0.8 | 0.5 | 0.8 | 0.4 |
| Zn | 0.2 | — | 0.2 | 0.08 | 0.2 |
| In | — | 0.007 | 0.06 | 0.001 | 0.5 |
| Cd | — | — | 1.0 | 1.4 | — |
| Cu | 2.8 | 1.6 | 1.2 | 1.2 | 1.2 |
| Sn | 8.8 | 3.3 | 11.0 | 18.2 | 18.1 |
| Pb | 2.2 | 2.1 | 4.8 | 8.1 | 9.6 |
| Ni | 0.002 | 0.02 | 0.03 | 0.001 | 0.001 |

## 4. Conclusions

Thus, we experimentally demonstrated a new high-temperature thermo-mechano-chemical method of PCB gasification by the detonation-born gasification agent composed of the ultra-superheated $H_2O$-$CO_2$ mixture. For the demonstration, a new experimental setup and accompanying measurements of gaseous and solid gasification products by chromatography–mass spectrometry, CHN analysis, wet laser diffraction, and X-ray fluorescence are used. The detonation-born gasification agent with a temperature exceeding 2000 °C is produced in the PDG operating on detonations of near-stoichiometric natural gas–oxygen mixture. The PDG produces pulsed supersonic jets of the gasifying agent and pulsed shock waves in the flow reactor with a batch of waste PCBs, thus exerting intense thermo-mechano-chemical action on the waste, leading to its fine grinding and gasification. The following main findings are worth emphasizing:

1. A high degree of gasification of organic matter in the flow reactor is achieved, which is over 98 wt%. Some solid mass (about 20 wt% of the processed PCBs) is removed from the flow reactor with the escaping gas in the form of very fine powder particles and is partly (about 10 wt%) trapped by the cyclones in the exhaust cleaning system. Metal inclusions of all visible sizes accumulate only in the flow reactor and are not detected in powder samples extracted from the cyclones. The gasification efficiency of the particles removed from the flow reactor with the escaping gas ranges from 76 to 91 wt%, i.e., the removed solid particles are not fully gasified. This problem can be eliminated, e.g., by using fine filters and/or the cascade of communicating flow reactors.

2. The full gasification of organic substances in a 1 kg batch of PCBs is attained by PDG operation at a frequency of 2 Hz during about 240 s with the overall consumption of a combustible natural gas–oxygen mixture of 1.4 kg. When the PDG operates at a frequency of 1 Hz, full gasification of organic substances in the PCBs requires more time (350 s) but less combustible mixture (about 1.0 kg). Thus, an operation frequency of 1 Hz is preferable for economic reasons but less preferable in terms of performance.

3. The gaseous products of PCB gasification are mainly composed of $CO_2$ (52.2 vol%), CO (25.1 vol%), $H_2$ (15.7 vol%), $N_2$ (3.4 vol%), and $CH_4$ (2.3 vol%) with the share of

flammable gas components reaching about 45 vol%. At the PDG operation frequency of 2 Hz, the gaseous products still contain some amounts of tar in terms of benzene and its homologs. The tar must disappear with the increase in the PDG operation frequency, i.e., to improve the environmental friendliness of PCB gasification products, it is necessary to increase the PDG operation frequency above 2 Hz.

4. The solid residues of PCB gasification are composed of fine powder with visible metal inclusions. With metal inclusions extracted, the powder is composed of particles possessing a size less than 300–400 μm, including a large fraction of sizes less than 100 μm. This is the result of the thermo-mechano-chemical action of strong pulsed shock waves emanating from the PDG.

5. After the extraction of metal inclusions, the powder obtained by the thermo-mechano-chemical gasification of waste PCBs contain Sn, Pb, Cu, Ni, Fe, In, Cd, Zn, Ca, Si, Al, Ti, Ni, and Cl. Among these substances, Sn (10–20 wt%), Pb (5–10 wt%), and Cu (up to 1.5 wt%) are detected in the maximum amounts. In the powder submitted for analysis, precious elements Ag, Au, and Pt are not detected, i.e., these elements (if any) remain in metallic inclusions.

Future work will be focused on improving the efficiency of the thermo-mechano-chemical processing of waste PCBs by (1) decreasing the amount of solid mass removed from the flow reactor with the escaping gas and (2) increasing the rate of conversion of condensable hydrocarbons in gaseous gasification products with an increase in the PDG operation frequency.

**Author Contributions:** Conceptualization, S.M.F.; methodology, S.M.F. and V.A.S.; formal analysis, S.M.F., V.A.S., A.S.S. and I.A.S.; investigation, S.M.F., V.A.S., A.S.S., I.A.S., F.S.F. and J.K.H.; resources, A.A.S. and V.E.S.; data curation, A.S.S. and I.A.S.; writing—original draft preparation, S.M.F.; writing—review and editing, S.M.F.; supervision, S.M.F.; project administration, S.M.F.; funding acquisition, S.M.F. All authors have read and agreed to the published version of the manuscript.

**Funding:** This research received no external funding.

**Institutional Review Board Statement:** Not applicable.

**Informed Consent Statement:** Not applicable.

**Data Availability Statement:** The data are available upon request.

**Conflicts of Interest:** The authors declare no conflicts of interest.

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
