# Peer review of "Thermo-Mechano-Chemical Processing of Printed Circuit Boards for Organic Fraction Removal"

_waste, doi:10.3390/waste2020009_

Round 1
Reviewer 1 Report
Comments and Suggestions for Authors
The actual manuscript presents apply a new method for recycling waste PCBs, which is referred to as the high-temperature thermo-mechanical gasification of PCBs by the detonation-born GA, that is the ultra-superheated mixture of H2O and CO2. Generally, the studies are interesting and can attract a wide readership. However, some following major concerns need to be solved:
1. Think about adding one or two sentences in the abstract that specifically state the motivation or research gap. What particular issues or queries is this study trying to answer?
2. The author describes the drawbacks of pyrometallurgy, hydrometallurgy, and incineration in the introduction, but does not mention the disadvantages of pyrolysis, which should be added.
3. The resolution of graphs is poor.
4. The size of the text in the figure should be consistent.
Author Response
We are grateful to the reviewer for valuable comments. We have made our best to follow all the comments. All changes in the revised manuscript are marked in yellow.
The actual manuscript presents apply a new method for recycling waste PCBs, which is referred to as the high-temperature thermo-mechanical gasification of PCBs by the detonation-born GA, that is the ultra-superheated mixture of H2O and CO2. Generally, the studies are interesting and can attract a wide readership. However, some following major concerns need to be solved:
- Think about adding one or two sentences in the abstract that specifically state the motivation or research gap. What particular issues or queries is this study trying to answer?
To address this comment, we have added the following two sentences to the Abstract:
“Printed circuit boards (PCBs) are the main components of e-waste. In order to reduce the negative impact of waste PCBs on human health and the environment, it must be properly disposed.”
- The author describes the drawbacks of pyrometallurgy, hydrometallurgy, and incineration in the introduction, but does not mention the disadvantages of pyrolysis, which should be added.
As a matter of fact, we mention pyrolysis in the Introduction section of the original manuscript. The main disadvantage of pyrolysis is the generation of hazardous condensable (tar) gaseous hydrocarbons. To address this comment, we have extended the existing sentence as follows:
“During low-temperature pyrolysis, the organic matter is decomposed to the syngas containing noncondensable and hazardous condensable (tar) gaseous hydrocarbons and phenolic tar.”
- The resolution of graphs is poor.
We have replotted some figure to improve resolution.
- The size of the text in the figure should be consistent.
We have replotted some figures to make the text size consistent.

Reviewer 2 Report
Comments and Suggestions for Authors
This work presents a novel approach to electronic waste management, reporting the results of an experimental analysis of an innovative technology for the valorization of the organic fraction of PCBs. The manuscript is well-organized and written. This paper could be accepted with some revisions. In addition, the authors are suggested to address the following comments in order to meet the requirements of the Journal.
- What type of PCB was used? Is it an FR-2 type, FR-4 type, or any other type? That should be mentioned, as it may affect the products of the process.
- Can the composition of the PCB sample be provided (in section 2)? Thanks to material balances, these data allow the reader to understand the process better.
- In Table 2, it would be appropriate to include at least one other significant number (like in Table 1).
- Mass percentages and volume percentages are often used in the text. It would be better to specify whether the authors are referring to one or the other (%vol. or %wt.).
- Figure 11 shows a case reported in Table 5. It is unnecessary to report even an example of an easily interpretable result. I suggest removing Figure 11 also to reduce the number of figures.
- The quality of some Figures should be improved (for example, some details in Figure 1).
- Table 7 presents the percentages of metals in the powders at the end of the process. Why are they so much inferior to the compositions usually found in PCBs (See, for example, Romano, P.; Ippolito, N.M.; Vegliò, F. Chemical Characterization of an ARDUINO®Board and Its Surface Mount Devices for the Evaluation of Their Intrinsic Economic Value. Processes 2023, 11, 1911.)? Since they are residues from the process, should they also be higher than average concentrations? Explain the process outputs better.
- Line 348 states, “In the powder submitted for analysis, precious elements Ag, Au, and Pt are not detected.” This result is obviously due to the analysis method used. Has the possibility of analyzing the concentrations of precious metals using other techniques been evaluated (for example, ICP - inductively coupled plasma spectroscopy)? This aspect would allow the reader to understand the importance of solid residue valorizing.
Author Response
We are grateful to the reviewer for valuable comments. We have made our best to follow all the comments. All changes in the revised manuscript are marked in green.
This work presents a novel approach to electronic waste management, reporting the results of an experimental analysis of an innovative technology for the valorization of the organic fraction of PCBs. The manuscript is well-organized and written. This paper could be accepted with some revisions. In addition, the authors are suggested to address the following comments in order to meet the requirements of the Journal.
1. What type of PCB was used? Is it an FR-2 type, FR-4 type, or any other type? That should be mentioned, as it may affect the products of the process.
We have used FR-2 type PCBs. To address this comment, we have added this information to the text related to Figure 3.
2. Can the composition of the PCB sample be provided (in section 2)? Thanks to material balances, these data allow the reader to understand the process better.
The objective of this study was to remove the organic compounds from the PCBs, which was fulfilled successfully. Unfortunately, we could not analyze the original PCBs on the elemental composition. To address this comment, we have added the following sentence to Section 2:
“The elemental composition of waste PCBs was not analyzed.”
3. In Table 2, it would be appropriate to include at least one other significant number (like in Table 1).
Done
4. Mass percentages and volume percentages are often used in the text. It would be better to specify whether the authors are referring to one or the other (%vol. or %wt.).
Done
5. Figure 11 shows a case reported in Table 5. It is unnecessary to report even an example of an easily interpretable result. I suggest removing Figure 11 also to reduce the number of figures.
Contrary to Table 5, Figure 11 shows 3 curves from different probes taken from one sample. It provides additional useful information regarding a high degree of powder uniformity. To avoid misunderstanding, we have added the following sentence to the caption of Figure 11:
“The curves correspond to three specimens taken from different places of a powder sample.”
6. The quality of some Figures should be improved (for example, some details in Figure 1).
Done.
7. Table 7 presents the percentages of metals in the powders at the end of the process. Why are they so much inferior to the compositions usually found in PCBs (See, for example, Romano, P.; Ippolito, N.M.; Vegliò, F. Chemical Characterization of an ARDUINO®Board and Its Surface Mount Devices for the Evaluation of Their Intrinsic Economic Value. Processes 2023, 11, 1911.)? Since they are residues from the process, should they also be higher than average concentrations? Explain the process outputs better.
It is written in the original text that analyses are made for the remaining powders after the manual extraction of all visible (macroscopic) metallic inclusions. This means that most of metals are extracted and the amounts in Table 7 are related only to some residues in the powder. To avoid misunderstanding, we have added the following sentence in relation to Table 7:
“Remind that most of metals are extracted prior to the analyses and therefore their amounts in Table 7 are related only to some residues in the powders.”
8. Line 348 states, “In the powder submitted for analysis, precious elements Ag, Au, and Pt are not detected.” This result is obviously due to the analysis method used. Has the possibility of analyzing the concentrations of precious metals using other techniques been evaluated (for example, ICP - inductively coupled plasma spectroscopy)? This aspect would allow the reader to understand the importance of solid residue valorizing.
As noticed in our response to the previous comment, all visible (macroscopic) metal inclusions were extracted prior to the analyses. Therefore, the indicated sentence relates only to the residual powders. The precious elements Ag, Au, and Pt can surely be present in the extracted metal inclusions. To avoid misunderstanding, we have added several more sentences to the text relating to Figure 10:
“As seen, some metal inclusions like body parts, which could contain precious metals inside, are deformed rather than destroyed. Obviously, to open such body parts one has to apply a larger-diameter PDG for generating stronger blast waves,“
and to Table 7:
“This means that these elements (if any) remain in extracted metallic inclusions.”
In addition, we have added some clarification to the conclusions.

Round 2
Reviewer 2 Report
Comments and Suggestions for Authors
The revision improved the quality of the manuscript. For this reason, I express a favorable opinion of its publication in the journal.